# A Contingent Valuation-Based Method to Valuate Ecosystem Services for a Proactive Planning and Management of Cork Oak Forests in Sardinia (Italy)

**Luigi La Riccia** [1,*] **, Vanessa Assumma** [2] **, Marta Carla Bottero** [1] **, Federico Dell'Anna** [1]
**and Angioletta Voghera** [1]

1   Interuniversity Department of Regional and Urban Studies and Planning, Politecnico di Torino,
    10125 Turin, Italy; marta.bottero@polito.it (M.C.B.); federico.dellanna@polito.it (F.D.);
    angioletta.voghera@polito.it (A.V.)
2   Department of Architecture, Università di Bologna, 40136 Bologna, Italy; vanessa.assumma@unibo.it
*   Correspondence: luigi.lariccia@polito.it

**Abstract:** The paper develops a model through a contingent valuation approach to support public authorities in the exploration and assessment of ecosystem services (ESs) generated by forest and woodlands (FOWLs). This approach is employed to the cork oak forests of the Sardinia region (Italy) due to their ability in the provision and regulation of cultural and recreational values to society. The paper describes the economic valuation of cultural ESs through the contingent valuation method (CVM) with the purpose to explore residents and tourists' willingness to pay (WTP) preferences towards conservation, valorisation, and the management of Goceano's cork oak forests in Sardinia. The approach may help retain suitable support for DMs, planners, technicians, and operators for a better understanding of the ESs' role in policy decisions, leading FOWLs towards a learning process between the environment, human beings, and landscape to promote and develop a proactive landscape and forest planning and management within the region.

**Keywords:** stated preferences; willingness to pay (WTP); ecosystem services (ESs); forest and woodlands (FOWLs); landscape assessment

## 1. Introduction

Over the last several decades, the environment and its components have become ever more transversal in policy decisions, especially those dealing with urban and territorial transformations. The increasing uncertainty and ambiguity due, on the one hand, to climate change (e.g., droughts, or run-off alterations) and, on the other hand, to man-made factors (e.g., unmanaged fires, lack of forest management, or abandonment of rural and inner areas) require a radical action for a trend reversal from recent worrying predictions [1]. The latest United Nations Conference on Climate Change held in Glasgow [2] stressed the urgency to reduce emissions to nought by 2050, limit the increase in temperatures, and also reduce deforestation by protecting and recovering ecosystems.

The forest and woodlands (FOWLs) are the keepers of habitats and microhabitats where autochthonous flora and fauna species live in. These produce biological energy through the ecological connectivity of the biotopes that compose an environmental system and interact with neighbouring systems across different scales [3–5]. The environmental system's health mirrors environmental quality, social well-being, landscape value, and the economic attractiveness of that territory. Each of these features are the components for indirectly measuring the resilience of that system [6–8]. The Sustainable Development Goals of "Building sustainable and resilient cities and communities", "Climate Action", and "Life on Land" (SDGs 11, 13 and 15) clarify the need for protection, recovery, and enhancement to ensure a more sustainable accessibility of terrestrial ecosystems with the purpose of

arresting soil degradation and losses of ecosystem services (ESs) [9,10]. According to the State of Europe's Forests [11], on average, 70% of European forests are publicly accessible, 6% are for public recreation, and an index of creativity density recorded within the forests equal to 16 annual visits per inhabitant. FOWLs are valuable and fragile subsystems; their maintenance is fundamental for conserving their value, adaptability, and resilience and for valorising the local economy, as well as guaranteeing the safety of employees, residents, and tourists. Even if these features are widely recognised today, they are insufficient to operationalise a worldwide common response to limit current and potential future losses. For example, Mediterranean forests were threatened in 2021 by numerous fire events, due to high peaks of temperature which were sometimes co-triggered by man's carelessness. In addition, the progressive abandonment of rural settlements due to job demand, remote geographic location, or difficult accessibility, increases the difficulty in managing FOWLs and therefore causes their degradation [12–15] and exposure to natural hazards [16,17].

In light of this scenario, the ESs by FOWLs should be well-conserved and managed for both present and next generations more than ever [16,18–23]. In fact, they play a very important role in the generation of multiple benefits to people, such as of cultural and recreational types [11]. Besides the fact that recreational activities can contribute to the economic growth and attractiveness of a territory, public bodies should bear in mind that these can play a supporting role in FOWL preservation, valorisation, and management. A sustainable management of recreational activities can minimise the "use and consumption" trend in compromising territorial and landscape characteristics (e.g., neglect, inexperience, vandalism, disturbances to wild life, or diffusion of allochthonous flora and fauna species, among others).

Over the last few decades, economists approached the field of ecological economics to explore the relationships between environmental assets (i.e., pure public goods) and their associated economic values [24]. The idea that an environmental asset can express both biophysical and economic values has been recently consolidated in the ES literature [25,26]. The estimation of the FOWL economic value through stated preference methods can help DMs, planners, technicians, and operators to better understand the relevance of implementing ESs within policy decisions, thus integrating FOWL heritage within the learning process between the environment, human beings, and landscape. As stated by the authors of [27], the valuation of forest ecosystem services is mainly motivated by factors such as incentives for forestry management programmes, or payment for ecosystem services (PES), or even discovering people's preferences and their willingness to pay/or accept compensation related to forest heritage [27–34].

This contribution is part of a research project conducted between the 2019 and 2020 by a large group of researchers from Politecnico di Torino (Angioletta Voghera—Scient. Coordinator, Luigi La Riccia, Vanessa Assumma, Maurizio Bocconcino, Marta Bottero, Davide Canone, Federico Dell'Anna, Stefano Ferraris, Gabriella Negrini, Emanuela Rebaudengo, Emma Salizzoni) and commissioned by the Agenzia Fo.Re.S.T.A.S. of the Sardinia Regional Authority. This project was aimed at evaluating the ESs supplied by the Goceano's cork oak landscapes in the central–northern part of Sardinia, focusing on biophysical and economic valuations and selecting specific ESs both on regional and local scales to evaluate and map the multifunctionality value expressed by the cork oak forests [35]. The ESs selected in this valuation are: (i) provisioning—cork production, forage production, biomass production; (ii) regulation—hydrogeological protection, carbon sequestration; (iii) cultural—identity values (for residents and tourists).

The biophysical valuation was developed and described in specific papers [36,37] with regard to provisioning and regulating ESs, whereas this paper focuses on the economic valuation of cultural ESs. The contingent valuation method (CVM) is employed with the aim to explore users' willingness to pay (WTP) with respect to the conservation, valorisation, and management of cork oak landscapes of Sardinia (Italy). The objective of the paper is to monetise the WTP of residents and tourists to safeguard the Goceano cork oak area; residents were asked the tax amount they would be willing to pay annually, while

tourists were asked about the one-off amount they would be willing to pay. The valuation approach by means of WTP made it possible to obtain values useful for determining the total economic value (TEV) of the ESs of cork forests in Sardinia.

Therefore, the paper has been structured into the following sections: Section 2 illustrates the study case and focuses on Goceano's cork oak forests; Section 3 is dedicated to the methodological aspects related to the recreational ESs and to the economic evaluation methods for the WTP estimation; Section 4 describes the CVM application; and Section 5 discusses the survey results and provides an estimation of the total economic value (TEV). The results of the study are discussed in Section 6. The last section reports final considerations on the usefulness of the methodology and provides future research perspectives.

## 2. Study Area: The Goceano Cork Oak Landscape

Cork oak is a Mediterranean autochthonous and spontaneous species that well-adapts to both summer and winter climate conditions, and thanks to its "resilience", the species can be up to a century old. The region of Sardinia (Italy), such as Portugal and Spain, is characterised by a great presence of cork oak forests (*Quercus suber*), thus becoming a structural factor of their landscape. Sardinian communities have benefitted from cork oak timber for centuries and employed it for the production and manufacturing of various products, spanning from building materials, bottling, clothing, and so on. The material derived is completely renewable and does not require the felling of the plant. The cork oak landscape is part of Sardinian cultural heritage in harvesting, extraction, and manufacturing processes as well as in the use and construction of ancient machinery.

In Italy, it is estimated that the area of cork oak forests spans up to 168,000 ha [38]. Most of these forests are located within Sardinia, where cork oak forests cover about 140,000 ha of land both as pure stands or wooded pastures.

In the Sardinian region, forests typify large portions of the landscape (particularly in the subregions of Marghine-Goceano, Gallura, Monte Acuto, Nuorese, Sulcis-Iglesiente, Montiferru, and Mandrolisai, many of which are classified as 'internal areas'), taking the form of both pure stands of cork oaks (around 80,000 ha) and wooded pastures (around 40,000 ha) (see Figures 1 and 2). These are highly productive landscapes and are characterised by strong identity values; indeed, Sardinia is historically the main producer of cork in Italy [39,40].

The Goceano forest complex particularly includes those of Anela, Fiorentini, and Monte Pisanu for a total area of 4800 ha. It is located within the Optimal Territorial Ambit no. 4 (i.e., Ambito Territoriale Ottimale) "SUT Goceano" of the Territorial Regional Plan of Sardinia (TRP) and includes nine municipalities: Anela, Bottidda, Benetutti, Bono, Bultei, Burgos, Esporlatu, and Illorai e Nule.

The study area is considered highly relevant for this experimentation, since cork oaks play an essential role in this delicate ecosystem. In fact, the Goceano forest complex has considerable potential for active forestry and pastoral management; it is characterised by the presence of cork oak forests that are among the most productive in terms of quality and quantity. The persistence of the three forests of traditional forage-pastoral landscapes in this territory, which are made up of open areas of woodlands with a prevalent zootechnical function, is of considerable interest. This ecosystem, which is very delicate if not properly managed, risks disappearing as a result of opposing phenomena, such as the progressive expansion of forests in areas that are scarcely used by livestock, together with the lack of cork regeneration in overburdened areas.

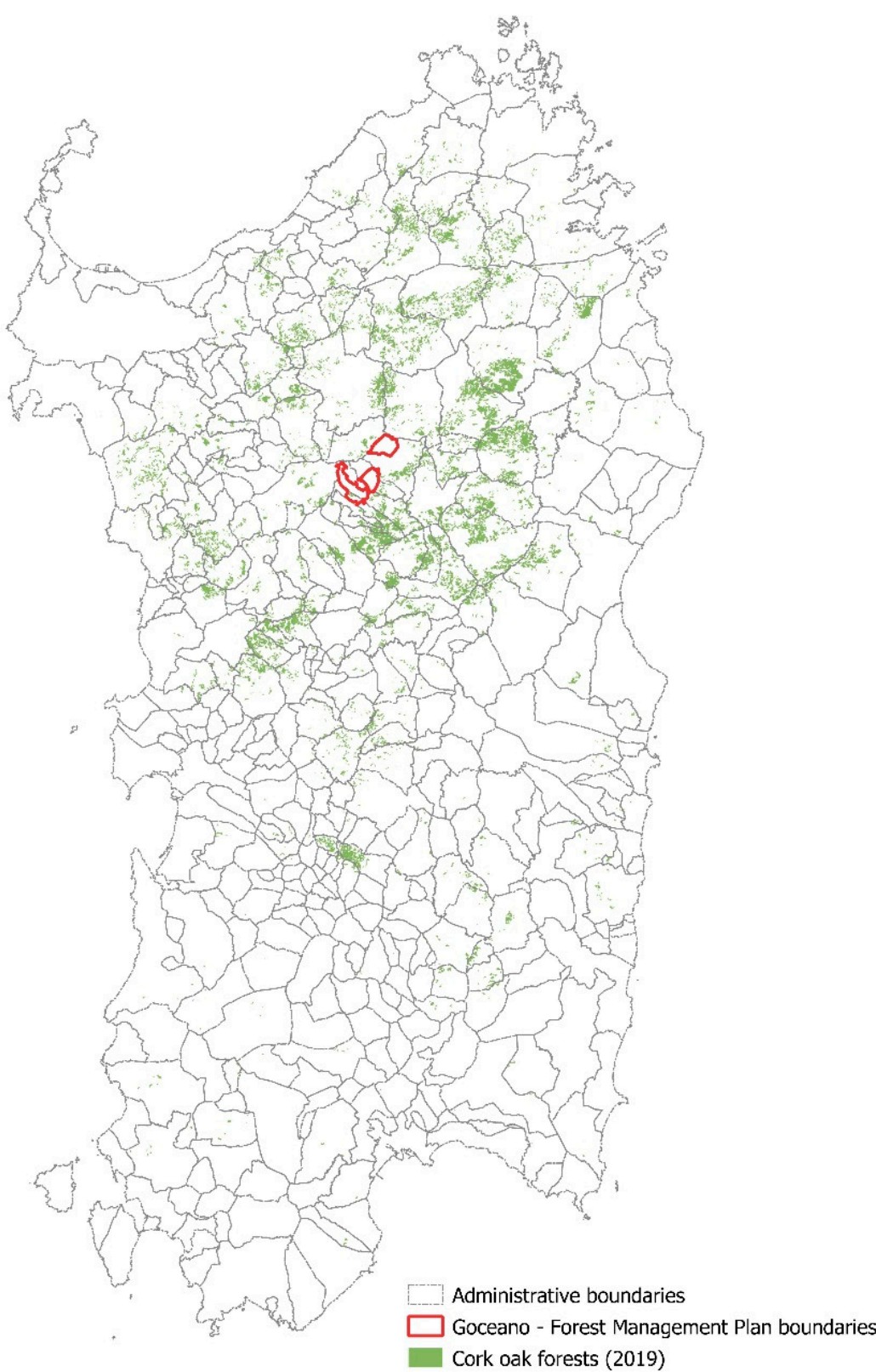

**Figure 1.** Distribution of cork oak forests in Sardinia. Elaboration: Luigi La Riccia and Angioletta Voghera, 2019.

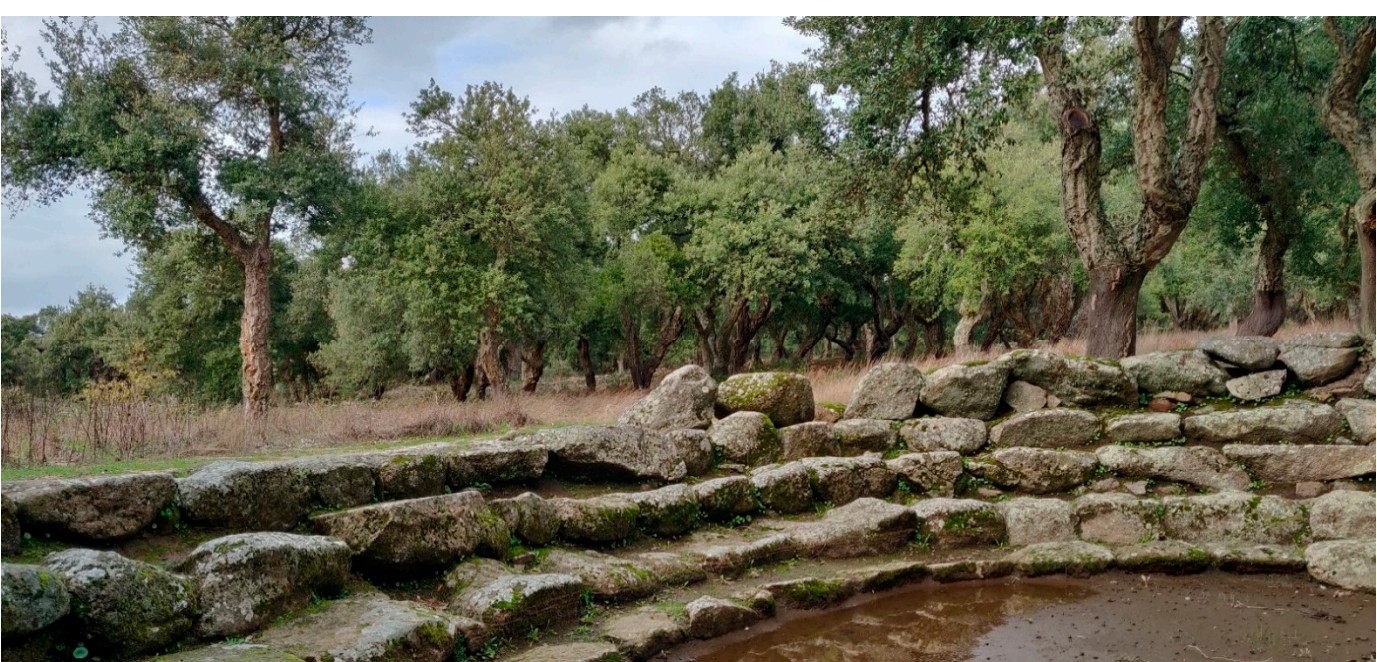

**Figure 2.** Cork oak forest landscape in Goceano: Sos Nibberos Protected Area, Monte Pisanu (photograph by Luigi La Riccia, 2019).

Cork oak forests represent one of the best examples of the close relationship between man and nature: forests with a high conservation value alternate with agricultural land, integrating extensive agriculture, forest grazing, hunting, and other recreational uses. In Sardinia, cork oak forests are traditionally multifunctional: they are agroforestry systems in which forest exploitation is almost always associated with grazing and agriculture. The relative weight of each component—forest, agriculture, and animal production—in the overall economic return of the system has changed over time. Recently, agriculture has been responsible for the opening up of large areas of forests, and cultivation in cork oak stands has been carried out extensively during the last century. Livestock, fed on natural vegetation and acorns or improved pastures, has been and still is one of the important products supported by cork oak stands. Other uses of cork oak forests are based on their rich biodiversity: mushroom picking, bee-keeping, and aromatic plants.

Current threats include increasing human pressures on environmental resources such as overgrazing and progressive deforestation, as well as land abandonment, resulting in poor forest management (bush encroachment and fires) caused by the spread of pests and diseases that lead to the decline of cork oak. These threats are generally caused by poor cork extraction and pruning practices that in many cases damage the regenerative tissues of plants, as well as by market competition and fluctuations in the price of cork [40]. These threats are also exacerbated by the effects of climate change.

The cork forest landscape, as mentioned, is a multifunctional landscape since the cork extraction activity never involves the elimination of the trees, but only their decortication (which consists of the separation of the bark from the trunk), which, if correctly performed, does not damage plants. This operation makes it possible to safeguard the biodiversity of these territories since these forests offer shelter to various species of animals, enriching ecosystems and providing them with ecosystem services of regulation, hydrogeological protection and carbon sequestration. These ecosystems are therefore highly resilient and, given the properties of the cork plant, they are also able to deal with the various biotic and abiotic disturbances due to risk factors, such as fires (since cork is essentially fireproof).

From an economic point of view, however, it is necessary to underline the critical issues due to competition on the international market of synthetic products (plastic caps) on cork products and other non-wood products, which are seriously endangering production.

According to [41], the fate of these 'traditional' landscapes depends heavily on innovative management efforts in this market. In this sense, the assessment of ecosystem services and the related mapping is absolutely essential to increase the knowledge of the value of these landscapes [42] and to define, through territorial planning and design, adequate enhancement perspectives complementary to those strictly economically productive.

Through the classification of forest areas differentiated by the level of density, thanks to the availability of data on dendrometric measurements, four forest density classes were therefore identified, together with the relative coefficients useful for calculating the supply and regulation services. The following table therefore shows the annual economic values relating to the supply and regulation SEs together with the total economic value (TEV) (see Table 1 and Figure 3). The integrated interpretation of the data has more substantially materialized the value of the multifunctionality of Sardinia's cork forests: the economic value associated with the production of cork is in fact very high and attests to the important productive function of these territories. It is true, however, that since productivity depends heavily on local trees, cork oaks do not significantly affect the possibility of simultaneously providing other ecosystem services of a more purely environmental nature, such as hydrogeological protection and carbon absorption. For this reason, the trade-offs typically existing between ecosystem services of supply and regulation [43], are more nuanced than in other contexts where the production of wood products prevails.

**Table 1.** Economic indicators of the ESs of the Goceano cork oaks and TEV (total economic value) percentage breakdown.

| ES | Economic Indicator | Estimation Method | Structure | Economic Value (EUR/year) | TEV (%) |
|---|---|---|---|---|---|
| Cork production | Market value of cork | Market price (EUR/q) | EUR/year | 58,879.15 | 40.2 |
| Fodder production | Market value of fodder | Market price (EUR/q) | EUR/year | 24,066.50 | 16.4 |
| Biomass production | Market value of biomass for energetic uses | Market price (EUR/q) | EUR/year | 24,034.26 | 16.4 |
| Hydrogeological protection | Surrogacy value of the protective function of forests | Surrogacy cost | EUR/year | 26,995.47 | 18.4 |
| Carbon sequestration | Market value of carbon | Market price (EUR/t) | EUR/year | 12,433.55 | 8.5 |
| | | | | 146,409.93 | 100% |

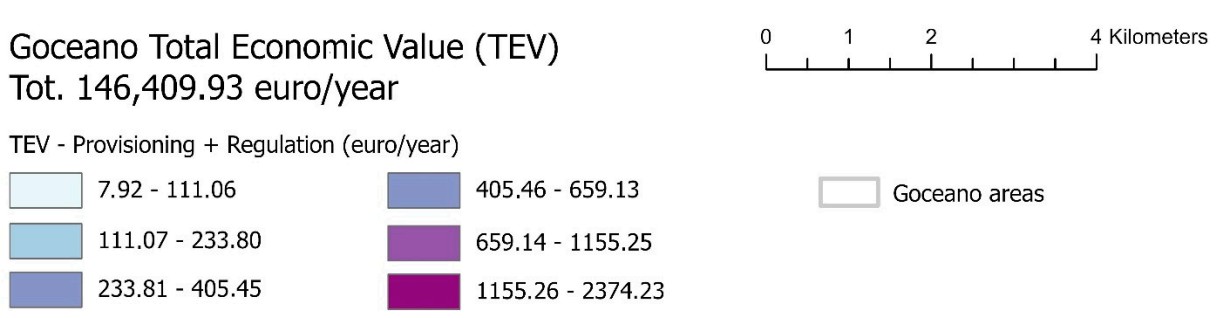

**Figure 3.** Total economic value of the provisioning and regulation ESs related to cork oak forests in a portion of Goceano. Elaboration: Luigi La Riccia and Angioletta Voghera, 2019.

### 3. Methodology

Before illustrating the application of the economic valuation, we refer to a general premise on the methods that are able to determine the economic value of environmental goods and services, which are considered useful by other authors.

In economics, goods and services are generally classified into:

- Private assets that are included in the market and regulated by buying and selling rules;
- Public assets that can be: (i) inseparable, because they are not divisible by simpler parts and are delivered to a specific user; (ii) non-competitive, since they are not dominated by market rules; and (iii) non-excludable, because everyone can equally access and use that asset or service [24,44].

For example, cork oak forests well-fit the category of public goods since they deliver multiple benefits to people, such as timber and biomass production, or cultural and recreational features (Table 2).

**Table 2.** Classification of goods in economics (adapted from [44]).

| Criteria | Excludability | Non-Excludability |
|---|---|---|
| Competitive | Private goods (e.g., cars, clothes) | Common goods (e.g., water) |
| Non-competitive | Club goods (e.g., works of art, cinema) | Public goods (e.g., forests, landscape) |

In the case of public goods, there are economic methods that help the analyst in building a hypothetical market in which it is possible to establish a monetary value for them by comparing the utility produced by goods with a decrease or increase in income [45].

The principle of TEV is widely recognised for public goods' valuation [46] since both tangible and intangible features are considered. In fact, TEV can be calculated by considering the use value and the not-use value. The first can be further subclassified into: (i) the direct use, meaning that those assets that can be extracted, consumed, or enjoyed (e.g., timber); (ii) indirect use, which is connected to the environment's functioning and services that have a positive effect on people who live nearby (e.g., recreational activities); and (iii) option value, which denotes that the utilisation of an asset for future benefits (e.g., individual's entertainment). The latter refers to intangible aspects, such as: (iv) bequest value, which refers to the value of leaving the asset optima to future generations (e.g., recreation for future generations) and (v) the existence value, to preserve a good by a potential damage or loss and also guaranteeing its inheritance for future generations (e.g., protected assets) [44].

Economists used to group economic valuation techniques into two broad categories:

- The monetary methods are based on monoparameter valuation to measure the benefits generated by a commodity or service. They can be employed with the purpose of stating or revealing preferences [47]. The "stated preference" methods can estimate users' preferences through the willingness to pay (WTP) or the willingness to accept for compensation (WTA), depending on whether the asset to be evaluated represents a positive or negative externality. The "revealed preferences" methods can valuate, for example, the indirect use of environmental and cultural assets by observing the information of private properties detected from real estate markets and that are indirectly connected to the characteristics of the public asset to be evaluated. For example, the hedonic prices method (HP) [48] can estimate the value of an environmental asset by considering a set of variables that influence the monetary values of nearby private properties [49,50]; or the travel cost method (TC) can calculate the expenses costs sustained by tourists for accessing public goods [51–53].
- The non-monetary methods can measure the value of environmental goods by considering individual characteristics and their globality as well. This is typical of composite

index valuation that employs a set of indicators that better represent the characteristics of an asset in order to provide its global performance, such as the economic value of landscapes [8,24,54,55]. Some research studies have recently added value to this stream by coupling non-monetary methods with mathematical modelling for a more dynamic interpretation of complex systems and thus facilitating the design recommendations capable of fostering transformations [56–58].

This paper is focused on the monetary category and more, in detail, on stated preference methods. It develops a CVM-based approach for the WTP estimation, which is intended as the maximum amount that an individual is willing to pay for having a good or use a service [59,60].

A literature review was developed by the authors using the Scopus database (https://www.scopus.com/, accessed on 14 November 2022) to select meaningful contributions in the relevant literature of stated preferences methods for the ESs' valuation and also to investigate their contribution in the field of forest and landscape planning and management:

"stated preferences" AND "WTP" AND "ecosystem services" = 35 results

"stated preferences" AND "WTP" AND "forest" = 30 results

"stated preferences" AND "forest" AND "landscape" = 20 results

"stated preferences" AND "WTP" AND "forest" AND "landscape" = 3 results

"stated preferences" AND "WTP" AND "ecosystem services" AND "forest" AND "landscape" = 2 results

Some publications were selected from the literature review because they are retained as significant for the objective of the research work (Table 3). For example, WTP can be estimated to elicit people's preferences on changes in the composition of forest trees [22], as well as on the structure and standing related to nature-based interventions. Ref. [61] focuses on landscape preferences in estimating WTP, whereas [62] deepens this aspect by considering the role of cultural ecosystem services. Ref. [63] employs the Delphi method in contingent valuation to assess WTP for preserving the Amazon rainforest by European households. Ref. [64] explores the WTP in the form of a donation for forest conservation and management. Ref. [65] has recently developed a choice modelling for exploring people's WTP as an ecosystem rehabilitation of a river basin and its ecosystem services.

**Table 3.** Selection of representative studies on the economic valuation of environmental goods and services.

| Author and Year | Description | Field of Application |
|---|---|---|
| Amirnejad et al., 2006 [66] | Existence value of Iranian forests through the CVM and dichotomous choice (DC). Use of the logit model to measure the individual WTP. | Ecological economics |
| Nielsen et al., 2007 [22] | Valuation of public preferences in forest recreational benefits and support of nature-based forest interventions. | Silviculture |
| Sayadi et al., 2009 [61] | Use of CVM and conjoint analysis to valuate landscape preferences and estimate the WTP for a landscape in Spain. | Rural development |
| Bastian et al., 2015 [62] | Estimation of the WTP for the appreciation of Saxony landscape in Germany and of its cultural ESs by tourists and visitors. | Landscape management |
| Tinch et al., 2015 [67] | Choice experiment valuation of changes in UK landscapes to explore the value associated with ES variations under different management regimes. Calculation of WTP off-site, on-site, and ex-post at two different time intervals (off-site). | Landscape management |

**Table 3.** *Cont.*

| Author and Year | Description | Field of Application |
|---|---|---|
| Cao et al., 2016 [68] | Exploration of influencing factors related to an urban ecosystem in China through logit and oprobit models, and estimation of the WTP for traveling to green spaces, forests, lakes, and rivers in Wenjiang (China). | Urban ecosystem and infrastructure |
| Price, 2017 [69] | Cost–benefit analysis (CBA) is combined with WTA/WTP for envisioning positive impact interventions. | Landscape economics |
| Navrud and Strand, 2018 [63] | Delphi method in CVM to measure the WTP by EU households for the protection of the Amazon rainforest. | Environmental protection |
| Schläpfer and Getzner, 2020 [70] | Empirical strategy based on choice experiment future management for the Austrian forests and investigation of the effects on WTP. | Forest management |
| Bamwesigye et al., 2020 [71] | Development of CVM for estimating the WTP for forest existence value in Uganda | Landscape management and planning |
| Alvarez et al., 2021 [72] | Estimation of the differences in WTP for urban and peri-urban forests in Florida (USA) by considering tree nativity, number of species, size of trees, and maintenance costs. | Urban forests |
| Hanim Mohd Sharif et al., 2021 [64] | Households' willingness to donate for the conservation and management of a recreational forest in Melaka using the double-bound CVM. | Forest management |
| Khan et al., 2022 [65] | Choice experiment to capture people's preferences for policy scenarios for vulnerable ecosystems. | Water management |

*The Contingent Valuation Method (CVM)*

The contingent valuation method (CVM) is a technique that is generally employed in the valuation of non-market assets and is based on real and potential users' preferences. CVM is employed to estimate the willingness to pay (WTP) as a monetary expression of people's preferences to preserve, improve, or simply access environmental and cultural resources, or the willingness to accept (WTA) compensation related to a modification of the asset value, or a renunciation of accessing it [59]. WTP/WTA estimation can help the evaluator to estimate the total economic value (TEV) of a given asset. The CVM can develop a fictitious market by capturing users' preferences and comparing the utility of a given asset, providing changes in their income without an effective monetary transaction [73]. The CVM can be synthesised according to the following steps:

1. The identification and description of the main characteristics of the asset to be valuated;
2. A representation of a hypothetical market and definition of payment modalities;
3. A selection of a homogeneous champion of the population who could be interested in using that asset;
4. The structuring of the survey addressed to the champion;
5. The implementation of the survey (e.g., questionnaire, interviews, and so on);
6. Survey data collection and elaboration;
7. The descriptive and inferential analysis of data;
8. An estimation of the TEV value.

CVM employment requires the development of a survey. The questionnaire is the most common form of users' engagement to know their preferences concerning a realistic scenario. In this way, the users' choices are led by the same motivations that govern their behaviour in a real market.

The structuring of the questionnaire is one of the crucial steps of this method. For example, the clarity in language, the level of detail in describing the asset, the specification

of use circumstances, or even how long and what type of payment is necessary for the asset use, can contribute to the survey reliability and help the evaluator in discovering users' preferences and determine who should belong to a homogeneous champion of the population [74].

There are different WTA/WTP elicitation formats:

1.  Open-ended questions: The users are asked to provide a value for WTP/WTA, without any prompting. Some typical questions provided in the questionnaire are:

    (a)　"How much would you be willing to pay for using the asset?"
    (b)　"How much would you be willing to pay for accepting the non-use of that asset?"

    Even though this method is the most popular, users could have some difficulty in autonomously providing a value, and there is a risk of them skipping questions which they may consider uncomfortable. Open-ended questions should be few but worthwhile.

2.  Closed-ended questions: The users are asked to provide their preferences by answering yes or no, or through an interval value of a monetary amount for paying or receiving compensation for that asset (e.g., from EUR 5 to EUR 100), or even an ordinal scale (e.g., "probably yes"), among others [74]. An example of a question could be: "Would you be willing to pay 10 EUR for the forests fire prevention programme?". Since close-ended questions are easier to be answered, these can help in the reduction of strategic answers.

3.  Iterative bidding questions: The interviewer initially provides a figure to the individual user. If they accept the figure, a higher figure is provided and the process is repeated until the user decides to stop it. Then, the interviewer proceeds to suggest reductions, until the respondent agrees to the reduced figure. This procedure appears to be the most frequently used. A good practice is to increase or decrease the value of the monetary amount (i.e., starting point) at the beginning of the questionnaire, for example, about twice the initial value. In the final phase, much smaller variations are preferable. The interviewer's skills are very important as they can contribute to the quality of the responses. The individual's ability to understand the declared amount is important for approaching the point of indifference for the interview and, subsequently, to appropriately reduce the variations. A limitation of this method is the production of alternate estimations (i.e., starting-point bias). In the case of a high initial amount, the individual tends to increase the WTP, whereas if the starting point is low, the user will tend to state a value lower than the current value.

4.  Dichotomous choice questions: This is an alternative approach to the iterative game, since the starting-point value can be varied randomly from one respondent to another, and the starting point coincides with the ending point [75].

5.  Payment card method: This allows the WTA/WTP of users to be identified by considering a set of monetary amounts concerning that asset (e.g., between EUR 5 and EUR 10, and more than EUR 10). Then, respondents are aided in providing more accurate answers by mirroring their maximum WTP/WTA [74]. However, it should be taken into account that the payment card method may imply an anchoring bias. The interviewer, after describing the asset to be valued and the hypothetical market, tries to identify the income class of the respondent. At that point, he/she explains the contents of the form corresponding to the interviewee's income category and, based on this, is asked to set a value for the asset being estimated. This elicitation format has recently been extended with more reliable variants (e.g., circular payment card—PC) [76,77].

A general structure of a questionnaire to be employed in the context of CVM should have: (i) an introductory section, containing general and attitudinal questions to determine the users' familiarity with the asset that is to be evaluated as well as his/her individual perceptions; (ii) a section containing questions to ask users' preferences in the form of WTA

or WTP, along with additional questions contributing to the consistency of the answers provided; and (iii) a last section, which is devoted to collecting users' socioeconomic information and can help in interpreting the results in expressing a given WTP/WTA with respect to other users. For example, the WTP can be different for subgroups of users (e.g., age, income, education, job, attitudinal preferences, and degree of attention in providing information with respect to cultural and environmental issues, among others).

The survey can be developed in various ways: by e-mail, telephone, online platforms, or through face-to-face interviews. The last modality is retained to be the most effective because it can provide the interviewer with detailed explanations and additional information about user preferences, even if it consumes considerable time and leads to resource losses. The online modality (e.g., LimeSurvey, Google Forms, or Survey Monkey, among others) can save time and resources, making the survey accessible to everyone and at any time.

Once the survey is concluded, the users' answers can be collected and organised in a Microsoft Excel environment to be subsequently processed through the use of probabilistic models, such as random utility models (RUM) or regression models [78–80].

It is possible to consider different methods for WTP/WTA elicitation. In the case of the open-ended response, simple elaborations of the WTP/WTA values can be developed, whereas in the case of the close-ended responses, statistical elaborations can take greater complexity.

The statistical models which are considered suitable for the estimation are those that can deal with discrete dependent variables, characterised by different specifications in the distribution of the error component. For example, WTP is considered in random utility models (RUM) as a random variable, whereby it is possible, by applying the different specifications, to estimate the most significant descriptive measures, such as the mean, median, and variance [73].

## 4. Survey Set-Up and Data Collection

The CVM was employed in the research project based on an exploratory approach and is finalised to provide decision makers an overview of the residents and tourists' willingness to pay (WTP) with regard to Goceano's cork oak landscape, and to orient the implementation of future policies in this territory.

The CVM is supported by a partial survey of stakeholders' preferences. This is due to the fact that it would not be possible to interview the entire population involved because it would increase the cost and time of the survey. The questionnaire design and data collection are reported below.

*Questionnaire Design*

The survey was conducted during 2019 (between July and September) and addressed to a sample of the population to assess, in an exploratory manner, the benefits delivered by the recreational ESs of cork oak forests. The questionnaire was administered to residents, tourists, and regional citizens, both online via Google Form and through face-to-face interviews in the Goceano's context and the regional territory, thanks to the synergic collaboration between the Politecnico di Torino and the Agenzia Forestas of the Sardinian region.

The questionnaire was structured into three sections, where the first section aims to detect the level of knowledge and perceptions about the environmental asset and its services, the second section provides a realistic scenario to determine the individual WTP, and the last section is devoted to the user's socioeconomic profile. The questions were structured according to the funnel technique (i.e., from simple questions that are easy to fill in, to those more specific). Open- and closed-ended questions were considered, as well as numerical preference scales (i.e., Likert scale).

The first section of the questionnaire is user-specific to detect the different points of view and perceptions in relation to cork oak forests:

1. Goceano's residents (those who live and work near cork oak forests and who have their own awareness of the identity value of the environmental asset) were asked, for instance, to indicate their city of residence, or how frequently they go to cork oak forests and the means of transportation used to reach them. Attention was paid to the local perception of their cultural and landscape values and the potential presence of eyesores. Moreover, the types of activity performed there were asked so as to detect the correlation between cork oak forests and recreational activities.

2. Tourists (those who travel for leisure and to visit Sardinia's environmental assets, e.g., Nuragic sites, cork oak forests, and traditional territories) were asked to identify their preferences in visiting cork oak forests as a tourism destination. For example, tourists were asked to provide the name of the places they were staying at (or would stay at) to obtain their degree of proximity to the Goceano's cork oak forests, how they came to know about these woods (e.g., tourist guides, suggestions from relatives and friends, organised trips, or the internet) and the factors which convinced them to go there (e.g., scenic, sport, art and culture, among others), their visiting duration, and the main elements which they considered important for this landscape.

3. Sardinian citizens were asked, for example, to specify their city of residence and the places within the region where they spent or would spend time at, and their reasons of choice. In addition, the respondents were asked whether they had ever visited cork oak forests and if so, in which area in Sardinia. Attention was paid to receptivity and accessibility features, asking about the place of stay, if any, and how they reached the cork oak forests.

Each questionnaire is structured with a scenario description for supporting the WTP elicitation. Below is an example concerning Sardinian citizens on the issue of forest fire risk:

*"Consider for a moment the current situation in Sardinia: the risk of forest fires, also increasing due to climate change, threatens the existence of the cork forest landscape. Let us suppose that public resources alone are not sufficient to manage the risk related to fires and a non-profit foundation takes on the task of conserving and safeguarding Sardinia's cork-oak forest heritage, such as restoring cork-oak vegetation, nature education activities and scientific research on cork-oaks. These objectives would only be achieved if enough people were willing to finance the foundation by donating a certain amount of money on a one-off basis. In your opinion, what should be the maximum amount of money (EUR) each person should donate to support this foundation for the management of the environmental good? (An only one value can be admitted)."*

The last section of the questionnaire collects socioeconomic information to reconstruct the user profile and of the whole champion. For example, classic questions on age group, level of education, occupation (if any), and income were considered in the questionnaire, and whether the anonymous respondent was a member of any non-profit environmental associations.

Table 4 shows the variables of the questionnaires, with an expected description and coding that are later used for the processing of the regressions using the Statistical Package for the Social Science software (SPSS 27, https://www.spss.it/, accessed on 11 July 2022).

**Table 4.** Variables of the CVM model.

| Variable | Description | Codification |
|----------|-------------|--------------|
| | Dependent Variable | |
| WTP [a] | Willingness to pay for the conservation and protection of the Goceano cork forests | In monetary terms (Euro) |
| | Independent Variables | |
| | Socioeconomic variables | |

**Table 4.** *Cont.*

| Variable | Description | Codification |
|---|---|---|
| AGE | Respondent's age group; 18–21, 22–24, 25–34, 35–44, 45–54, >55 | Individual choice of age group |
| GEN | Respondent's gender | 1 for male, 0 for female |
| EDU | Respondent's education level | Amount of school years |
| AFFIL | Respondent's affiliation to non-profit environmental associations | 1 for membership, 0 for non-membership |
| | Respondent's occupation | |
| WORK_STUD | Respondent is a student | 1 representing that the respondent is a student, 0 otherwise |
| WORK_FARM | Respondent is a farmer/craftsman/merchant | 1 representing that the respondent is a farmer, craftsman, or merchant, 0 otherwise. |
| WORK_ENTREP | Respondent is an entrepreneur | 1 representing that the respondent is an entrepreneur, 0 otherwise |
| WORK_DEALER | Respondent is a dealer | 1 representing that the respondent is a dealer, 0 otherwise |
| WORK_PROFES | Respondent is self-employed | 1 representing that the respondent is self-employed, 0 otherwise |
| WORK_RETIRED | Respondent is retired | 1 representing that the respondent is retired, 0 otherwise |
| | Reason why the respondent visited the Goceano cork oak forests | |
| MOTIVE_SCENIC | Scenic landscape | |
| MOTIVE_CULTURE | Art and culture | 1 indicates reason for visit, 0 indicates no reason |
| MOTIVE_SPORT | Sports and outdoor activities | |
| MOTIVE_OTHER | Other reasons | |
| | Activities generally carried out in cork oak forests [b] | |
| ACTIVE_WALK | Walk | |
| ACTIVE_LANDM | Land maintenance and management | |
| ACTIVE_FOOD | Food and wine | 1 indicates activity carried out, 0 indicates no activity |
| ACTIVE_RELAX | Relaxation | |
| ACTIVE_SPORT | Sport | |
| ACTIVE_OTHER | Other | |
| | Landscape elements valued and to be enhanced | |
| LANDSC_MAN | Human–environment coexistence | |
| LANDSC_RECREAT | Recreational aspect | |
| LANDSC_WOOD | Ancient trades in the forest | 0 representing no interest and 1 full interest in landscape element |
| LANDSC_SMELL | Olfactory aspect | |
| LANDSC_FOOD | Food and wine aspect and sylvan pastoral context | |
| LANDSC_SPIRIT | Spiritual/religious aspect | |
| | Means of transport used to reach the Goceano cork oaks | |
| TRANSP_FOOT | On foot | |
| TRANSP_BICYCLE | By bicycle | 1 indicates used means of transport, 0 indicates unused means of transport |
| TRANSP_CAR | By car | |
| TRANSP_OTHER | Other means | |

**Table 4.** *Cont.*

| Variable | Description | Codification |
|---|---|---|
| | Knowledge of the existence of the Goceano cork forests [c] | |
| MEAN_GUIDES | Consulting tourist guides | |
| MEAN_RELATIVE | Relying on organised trips | 1 indicates how it became known, 0 means not used |
| MEAN_INTERNET | Surfing the internet | |

[a] WTP is expressed as an annual payment for residents. While for tourists, it is expressed as a one-off payment.
[b] For residents only. [c] For tourists only.

## 5. Survey Results

In total, 100 anonymous questionnaires were collected (80% response rate) by face-to-face interviews, but due to incomplete answers, only 78 questionnaires were considered valid; 32 for residents, 46 for tourists.

### 5.1. Descriptive Analysis of the Sample

The main socioeconomic data of respondents are shown in Table 5. The frequency analysis reveals an equal distribution of people under and over 45 years old (46.9% of the sample between 18 and 44 years old) for residents. The educational profile indicates that more than 90% of respondents have at least a higher education. Respondents' travel attitudes and tourism-related environmental awareness are summarised in Table 6. About half of the resident respondents visit the Goceano cork forests for work (46.9%). The rest of the resident respondents for scenic (34.4%), cultural (9.4%), and sporting reasons (25%). This result testifies to the fact that cork oak forests are frequented mainly by workers, rather than by residents for recreational activities. With regard to recreational activities carried out by residents (Table 7) within the cork forest, the most frequent are walking (34.4%), relaxation (25%), and sports (18.8%). A total of 75% of the respondents stated that they reach the park by car (Table 8). When asked which elements of the cork oak landscape they most appreciated and considered important to enhance (Table 9), the aspect related to human–environment coexistence was the most important (65.9%). The visual aspect follows (53.1%). The organisation of excursions and rest points is also an important aspect (37.5%). The aspect related to ancient forest trades and olfactory followed (18.8% and 15.6%, respectively).

**Table 5.** Socioeconomic data of respondents.

| Age | | 18–24 | 25–34 | 35–44 | 45–54 | >54 |
|---|---|---|---|---|---|---|
| Residents | Freq. (%) | 4 (12.5) | 3 (9.4) | 8 (25) | 11 (34.4) | 6 (18.8) |
| Tourists | Freq. (%) | 4 (8.7) | 5 (10.9) | 4 (8.7) | 6 (13) | 11 (23.9) 16 (34.9) |
| GEN | | Male | | Female | | |
| Residents | Freq. (%) | 23 (71.9) | | 9 (28.1) | | |
| Tourists | Freq. (%) | 29 (63) | | 17 (37) | | |
| EDU | | No qualification | Primary school | Secondary school | High school graduate | University degree |
| Residents | Freq. (%) | 0 (0) | 2 (6.3) | 11 (34.4) | 16 (50) | 3 (9.4) |
| Tourists | Freq. (%) | 5 (10.9) | 0 (0) | 12 (26.1) | 22 (47.8) | 7 (15.2) |

**Table 6.** Reasons for residents and tourists to visit the cork oak forests.

| Reason | | Motive_Scenic | Motive_Culture | Motive_Sport | Motive_Work |
|---|---|---|---|---|---|
| Residents | Freq. (%) | 11 (34.4) | 3 (9.4) | 8 (25) | 15 (46.9) |
| Tourists | Freq. (%) | 28 (60.9) | 6 (13) | 11 (23.9) | 0 (0) |

**Table 7.** Main activities carried out by residents.

| Activity | | Active_Walk | Active_Landm | Active_Food | Active_Relax | Active_Sport | Active_Other |
|---|---|---|---|---|---|---|---|
| Residents | Freq. (%) | 11 (34.4) | 1 (3.1) | 0 (0) | 8 (25) | 6 (18.8) | 9 (28.1) |

**Table 8.** Means of transport used to reach the site in question.

| Transport Means | | Transp_Foot | Transp_Bicycle | Transp_Car | Transp_Other |
|---|---|---|---|---|---|
| Residents | Freq. (%) | 2 (6.3) | 2 (6.3) | 24 (75) | 1 (3.1) |
| Tourists | Freq. (%) | 9 (19.6) | 5 (10.9) | 30 (65.2) | 10 (21.7) |

**Table 9.** Landscape elements and disturbances felt by respondents.

| Landscape Elements | | Land_Visual | Land_Man | Land_Recreat | Land_Wood | Land_Smell | Land_Food | Land_Spirit |
|---|---|---|---|---|---|---|---|---|
| Residents | Freq. (%) | 17 (53.1) | 21 (65.9) | 12 (37.5) | 6 (18.8) | 5 (15.6) | 2 (6.3) | 3 (9.4) |
| Tourists | Freq. (%) | 22 (47.8) | 14 (30.4) | 24 (52.2) | 12 (26.1) | 4 (8.7) | 11 (23.9) | 3 (6.5) |
| ENVIRONMENTAL DISTURBANCES | | Yes | | No | | | | |
| Residents | Freq. (%) | 7 (21.9) | | 25 (78.1) | | | | |
| GRAZING ACTIVITIES | | Yes | | No | | | | |
| Residents | Freq. (%) | 6 (18.8) | | 26 (81.3) | | | | |

## 5.2. Aggregating and Interpreting WTP

The objective of the paper was to monetise the WTP of residents and tourists to safeguard the Goceano cork oak area. Resident respondents were asked the amount they would be willing to pay annually as a tax and the results obtained are statistically described in Table 10. Tourists were asked about the one-off amount they would be willing to pay and the results obtained are also statistically described in Table 10.

**Table 10.** WTP stated by residents and tourists for preserving the area.

| WTP | | Mean (EUR/Year) | SD (EUR/Year EUR) | Median (EUR/Year) | Mode (EUR/Year EUR) | Zero-Bids (%) | Min. (EUR/Year EUR) | Max. (EUR/Year EUR) | N. (-) |
|---|---|---|---|---|---|---|---|---|---|
| Residents (whole sample) | Freq. (%) | 11.78 | 25.17 | 1 | 0 | 16 (50) | 0 | 100 | 32 |
| Residents (positive WTP) | Freq. (%) | 23.56 | 31.829 | 10 | 10 | 0 (0) | 2 | 100 | 16 |
| | | Mean (EUR) | SD (EUR) | Median (EUR) | Mode (EUR) | Zero-Bids (%) | Min. (EUR) | Max. (EUR) | N. (-) |
| Tourists (whole sample) | Freq. (%) | 17.57 | 21.97 | 10 | 10 | 1 (2) | 0 | 100 | 46 |
| Tourists (positive WTP) | Freq. (%) | 17.96 | 22.06 | 10 | 10 | 0 (0) | 1 | 100 | 45 |

Focusing on residents, 50% of the respondents (N = 16) declared a WTP of EUR 0. This result may be due to the fact that about 50% of the sample consists of personnel employed in forest management and maintenance activities. On the other hand, it can be said that residents recognise the site as a public good to be enjoyed free of charge. Considering the

sample of full residents, the average WTP is 11.78 EUR/year. Instead, tourists declared a higher WTP, recognising the recreational and cultural value of this natural heritage. The average WTP stands at EUR 17.57 to enjoy the cork oak forests. However, it must be remembered that WTP is a sum of money that should be paid annually, which is why it is lower than that of tourists. The different answers of the respondent sample do not allow a direct comparison of results to identify the overall WTP. In order to be able to aggregate the WTP of residents to that of tourists, a reference was made to the fact that the sum declared by the former is a constant financial performance that occurs at annual intervals and that in order to be able to calculate a total value per resident, it is necessary to anticipate them at the time of estimation by means of the formula for calculating initial accumulation. In particular, a benefit duration of 25 years, equal to the time between generations, and a discount rate of 3% were considered. The estimate resulted in a total WTP per resident of about EUR 205.

### 5.3. Estimation Results

The econometric estimation models developed in this research provided insight into the associations between the respondents and their WTP. This information complements and enriches the understanding of the main investigative problem of this research, namely the assessment of WTP. The statistical technique used in this study is a multivariate analysis by means of a linear multiple regression analysis for each subgroup identified, taking WTP into consideration as the dependent variable.

A first regression considered all variables to test their significance. In detail, the *p*-value of each variable was taken into account to select the variables to be included in a reduced model. Considering variables with *p*-value < 1%, the reduced model is shown in Table 11. From the results of the reduced model, those who are older are willing to pay less (bAGE = −2.388). This relationship may be due to the type of activities carried out in the forest, perhaps more in line with the habits of young people. In fact, the park is located close to a campsite, which is a very common arrangement among young people. Those with a higher level of education are willing to pay more (bEDU = 0.556). This is likely because a higher income often correlates with an awareness of the ecosystem services provided by the cork oak forest. The entrepreneurs are more willing to pay for the conservation of the area than the others, probably due to the fact that they have a higher income (bWORK_ENTREP = 82.044). Those who have participated in non-profit environmental associations are willing to pay more (bAFFIL = 6.811). This result is expected, as the expressed WTP is influenced by the sensitivity of the respondents. Those who go to the forest more often are willing to pay less (bFREQ = −2.439). This result is probably due to the fact that those who go most are workers in the forest. Referring to the reasons why respondents go to the forest, those who go for cultural reasons are willing to pay less (bMOTIVE_CULTURE = −71.139), while those who benefit from the scenic benefit would be willing to pay more (bMOTIVE_SCENIC = 9.861). The spiritual value of the park is most likely a motivation for visitors to go to the forest. Those who go to the forest to walk are willing to pay more (bACTIVE_WALK = 29.356). The area in question could be one of the areas available for this activity in the surrounding area. Those who manage and maintain the park are willing to pay (bACTIVE_LANDM = −65,670). Respondents seem to be willing to pay more for the elements of the cork oak landscape that refer to the promotion of ancient forest crafts (bLAND_WOOD = 17.691) and the olfactory aspect (bLAND_SMELL = 44.378). Those who noticed elements of environmental disturbance (e.g., visual or acoustic disturbance) in the cork oak forests are willing to pay less (bDIST = −12.997), whereas those who consider pasteurisation as a characteristic element of the Goceano landscape are willing to pay more (bPAST = 5.453). In the restricted model that can be considered reliable, variables are significant and have a correct sign in line with the expected sign.

**Table 11.** Econometric analysis of the sample of residents.

| Variables | Non-Standardised Coefficients | | | | 95.0% Confidence Interval for b | |
|---|---|---|---|---|---|---|
| | b | Standard Error | t | *p*-Value | Lower Limit | Upper Limit |
| Constant | 9.262 | 1.978 | 4.682 | 0.001 | 4.786 | 13.737 |
| Socioeconomic variables | | | | | | |
| AGE | −2.388 | 0.229 | −10.427 | 0.000 | −2.907 | −1.870 |
| EDU | 0.556 | 0.110 | 5.044 | 0.001 | 0.307 | 0.806 |
| WORK_STUD | −33.575 | 2.190 | −15.332 | 0.000 | −38.529 | −28.621 |
| WORK_EMPLOY | −11.965 | 1.625 | −7.364 | 0.000 | −15.641 | −8.289 |
| WORK_ENTREP | 82.044 | 2.426 | 33.816 | 0.000 | 76.555 | 87.532 |
| WORK_DEALER | −36.672 | 1.969 | −18.624 | 0.000 | −41.126 | −32.218 |
| WORK_PROFES | −62.157 | 1.940 | −32.037 | 0.000 | −66.545 | −57.768 |
| Environmental activities and visiting attitude | | | | | | |
| AFFIL | 6.811 | 1.169 | 5.824 | 0.000 | 4.166 | 9.456 |
| FREQ | −2.439 | 0.263 | −9.288 | 0.000 | −3.033 | −1.845 |
| MOTIVE_SCENIC | 9.861 | 1.267 | 7.786 | 0.000 | 6.996 | 12.726 |
| MOTIVE_CULTURE | −71.139 | 2.488 | −28.589 | 0.000 | −76.767 | −65.510 |
| TRANSP_FOOT | 11.528 | 2.296 | 5.022 | 0.001 | 6.335 | 16.721 |
| TRANSP_CAR | 3.010 | 0.809 | 3.720 | 0.005 | 1.180 | 4.841 |
| TRANSP_BICYCLE | 15.587 | 1.479 | 10.540 | 0.000 | 12.241 | 18.932 |
| ACTIVE_WALK | 29.356 | 2.031 | 14.457 | 0.000 | 24.762 | 33.949 |
| ACTIVE_LANDM | −65.670 | 2.164 | −30.342 | 0.000 | −70.566 | −60.774 |
| ACTIVE_RELAX | −5.586 | 1.005 | −5.559 | 0.000 | −7.859 | −3.313 |
| LAND_WOOD | 17.691 | 2.179 | 8.118 | 0.000 | 12.762 | 22.621 |
| LAND_SMELL | 44.378 | 1.419 | 31.282 | 0.000 | 41.169 | 47.587 |
| LAND_FOOD | −2.815 | 1.097 | −2.566 | 0.030 | −5.296 | −0.333 |
| DIST | −12.997 | 1.993 | −6.521 | 0.000 | −17.506 | −8.488 |
| PAST | 5.453 | 0.923 | 5.909 | 0.000 | 3.365 | 7.541 |
| F-value | 854.883 | | | | | |
| *p*-value | 0.000 | | | | | |
| $R^2$ | 0.998 | | | | | |

Considering the tourists' answers, the following results were obtained (Table 12). Older people declared a higher WTP (AGE = 3.811). Tourists' WTP increases with increasing years of study (EDU = 1.875). Employees, pensioners, and professionals are more willing to pay more. Those who recognise a cultural value of the property declared a higher WTP (MOTICE_CULTURE = 12.69). Those who stay in accommodations such as BnBs pay more, likely related to economic conditions. Those who recognise recreational and food values are willing to pay more for its preservation.

**Table 12.** Econometric analysis of the sample of tourists.

| | Non-Standardised Coefficients | | t | p-Value | 95.0% Confidence Interval for b | |
|---|---|---|---|---|---|---|
| | b | Standard Error | | | b | Standard Error |
| (Constant) | 136.854 | 31.626 | 4.327 | 0.000 | 71.431 | 202.278 |
| Socioeconomic variables | | | | | | |
| AGE | 3.811 | 1.883 | 2.024 | 0.055 | −0.084 | 7.706 |
| GEN | −17.026 | 5.562 | −3.061 | 0.006 | −28.531 | −5.521 |
| EDU | 1.875 | 0.620 | 3.026 | 0.006 | 0.593 | 3.157 |
| WORK_EMPLOY | 22.703 | 9.014 | 2.519 | 0.019 | 4.056 | 41.351 |
| WORK_RETIRED | 31.878 | 7.888 | 4.041 | 0.001 | 15.560 | 48.196 |
| WORK_PROFES | 24.250 | 9.540 | 2.542 | 0.018 | 4.516 | 43.985 |
| Environmental activities and visiting attitude | | | | | | |
| MEAN_GUIDES | −46.486 | 18.759 | −2.478 | 0.021 | −85.292 | −7.680 |
| MEAN_RELATIVE | −23.076 | 7.081 | −3.259 | 0.003 | −37.724 | −8.428 |
| MEAN_INTERNET | −25.320 | 10.549 | −2.400 | 0.025 | −47.141 | −3.498 |
| MOTIVE_CULTURE | 12.690 | 6.948 | 1.826 | 0.081 | −1.683 | 27.064 |
| ACCOM_OTHER | −58.438 | 17.700 | −3.302 | 0.003 | −95.053 | −21.824 |
| ACCOM_CAMP | −23.761 | 9.311 | −2.552 | 0.018 | −43.023 | −4.499 |
| ACCOM_BNB | 37.750 | 14.889 | 2.535 | 0.018 | 6.949 | 68.551 |
| ACCOM_RELATIVE | 13.541 | 7.013 | 1.931 | 0.066 | −0.966 | 28.048 |
| TRANSP_OTHER | −97.447 | 24.849 | −3.922 | 0.001 | −148.851 | −46.044 |
| TRANSP_FOOT | −32.435 | 8.201 | −3.955 | 0.001 | −49.401 | −15.469 |
| TRANSP_BICYCLE | −85.747 | 26.169 | −3.277 | 0.003 | −139.881 | −31.612 |
| TRANSP_CAR | −124.405 | 25.629 | −4.854 | 0.000 | −177.422 | −71.388 |
| TIME_ONEDAY | −25.347 | 6.258 | −4.050 | 0.000 | −38.293 | −12.401 |
| RETURN_YES | −23.180 | 7.547 | −3.071 | 0.005 | −38.791 | −7.568 |
| LANDSC_RECREAT | 13.650 | 4.874 | 2.800 | 0.010 | 3.567 | 23.733 |
| LANDSC_FOOD | 39.956 | 5.766 | 6.930 | 0.000 | 28.029 | 51.883 |
| F-value | 5.486 | | | | | |
| p-value | 0.000 | | | | | |
| R² | 0.840 | | | | | |

## 5.4. Estimation of the TEV

Table 13 shows the value of the individual WTP for residents and tourists and the overall WTP. The individual WTP was obtained by multiplying the number of residents/tourist arrivals by the respective WTP obtained through the regression model. The data taken into account for the calculation relate to the year 2019, which is when the survey for this study was conducted, prior to the pandemic, which, as we are all aware, disrupted tourism flows, if not cancelled, for reasonable cause.

**Table 13.** Calculation of the individual WTP of residents, the individual WTP of tourists and the overall WTP mean per Goceano's municipalities.

| Goceano's Municipalities | Area (km²) | Resident Individual WTP (Entire Life) | Residents 2019 * (No.) | Residents Individual WTP | Tourism Arrives ** | Tourism Individual WTP | Tourists Individual WTP (EUR) | Overall WTP (EUR) |
|---|---|---|---|---|---|---|---|---|
| Anela | 36.89 | | 620 | (EUR) | 2019 (no.) | | 0 | 7304 |
| Benetutti | 94.45 | | 620 | 127,100 | 0 | | 0 | 127,100 |
| Bono | 74.54 | | 1809 | 370,845 | 729 | | 12,809 | 383,654 |
| Bottidda | 33.71 | | 3481 | 713,605 | 128 | | 2249 | 715,854 |
| Bultei | 96.83 | 205 | 673 | 137,965 | 112 | 17.57 | 1968 | 139,933 |
| Burgos | 18.08 | | 897 | 183,885 | 35 | | 615 | 184,500 |
| Esporlatu | 18.4 | | 899 | 184,295 | 0 | | 0 | 184,295 |
| Illorai | 57.19 | | 382 | 78,310 | 0 | | 0 | 78,310 |
| Nule | 51.95 | | 830 | 170,150 | 0 | | 0 | 170,150 |
| Total | 482.04 | - | 1365 | 279,825 | 0 | - | 0 | 279,825 |
| Mid. value | - | | 10,956 | 2,245,980 | 1004 | - | 17,640.28 | 2,263,620 |

* ISTAT—Atlante statistico dei Comuni 2019 https://asc.istat.it/ASC/, accessed on 17 July 2022. ** Notes: Tourism arrives in proximity of Goceano's cork forests. http://osservatorio.sardegnaturismo.it/it/dashboard/dati-2019, accessed on 11 July 2022 (SIRED, Assessorato del Turismo della Sardegna).

For instance, among the municipalities taken into consideration, the municipality of Bono has the highest individual WTP relative to residents (41,006 euros), followed by the municipality of Benetutti (21,310 euros), whereas the municipality of Esporlatu has the lowest individual WTP relative to residents (4500 euros), likely as a result of the municipality's small population (only 382). Regarding the individual WTP of tourists near cork oak forests, the 2019 visitor movements made available by the Region of Sardinia's Tourism Department were considered. There are certain tourism flows that are not reported because of a lack of tourism accommodation and facilities, or the number of arrivals was much too low that tourism observatories made them unavailable. Due to the lack of data, the number of tourist arrivals for the municipalities concerned was assumed to be zero (i.e., Anela, Burgos, Esporlatu, Illorai, and Nule). In order to obtain the overall WTP, the total WTP of locals and tourists have been summed up. The WTP total sum for the Goceano area is EUR 2,263,620.

The overall value of WTP obtained by summarizing the total WTP for residents and tourists (Table 14) contributes to the final calculation of TEV (Table 15), which is equal to EUR 2,410,030, and thus a monetary valuation of cork oak forests that holds together the ecosystem and cultural-recreational value is obtained.

**Table 14.** Calculation of the overall WTP mean related to the Goceano's surface area (km²).

| WTP Residents (EUR) | WTP Tourists (EUR) | Overall WTP (EUR) | Overall WTP (EUR/km²) |
|---|---|---|---|
| 2,245,980 | 17,640 | 2,263,620 | 4696 |

**Table 15.** Calculation of the final TEV that takes into account both the ecosystemic and cultural-recreative results.

| Goceano's Cork Oak Forests Surface (ha) | TEV Ecosystemic | TEV Cultural-Recreative | Overall TEV | Cork Oak Forests Parametric Value (EUR/ha) |
|---|---|---|---|---|
| 4800 | 146,410 | 2,263,620 | 2,410,030 | 502 |

## 6. Discussion

The evaluation carried out can be compared with other studies relevant in the scientific literature. Particularly, the selection of article proposed by the authors of [27] are CVM that cover the time range 2006–2022 and are employed to support forest conservation, management, and restoration. This selection has facilitated the authors in the comparison between their WTP annual mean (USD) and the one obtained for the Goceano's cork oak forests. Thus, Table 16 below validates the results:

**Table 16.** Comparison of the annual mean WTP value with existing contingent valuation studies. (adapted from [27]).

| Authors and Year | Description | Country | Annual Mean WTP Value (USD) |
|---|---|---|---|
| Amirnejad et al. (2006) [66] | Estimation of the existence value of forests | Iran | 44.39 |
| Adams et al. (2008) [81] | Conservation of natural protected areas | Brazil | 1.65 |
| Chukwuone and Okorji (2008) [82] | Community forests management for conservation of non-timber forest products | Nigeria | 6.53 |
| Sattout et al. (2007) [80] | Economic valuation of cedar relics | Lebanon | 63.95 |
| Tao et al. (2012) [83] | Valuation of forest ecosystem services | China | 46.16 |
| Dumenu (2013) [84] | Economic valuation of urban forests | Ghana | 27.17–28 |
| Ansong and Røskaft (2014) [85] | WTP estimation for sustainable forest management | Ghana | 11.73–24.02 |
| Arowolo et al. (2014) [86] | WTP valuation for sustainable management of community forests | Nigeria | 37 |
| Tuan et al. (2014) [87] | WTP estimation for forest restoration | Vietnam | 7.47–8.32 |
| Al-Assaf (2015) [88] | Economic valuation of forest services | Jordan | 22.40 |
| Amiri et al. (2015) [89] | Valuation of conservation value of myrtle forests | Iran | 22.40 |
| Chen (2015) [90] | WTP for the conservation of urban heritage trees | China | 4.71–5.96 |
| Dare et al. (2015) [91] | Management of urban trees forest | Nigeria | 32.80 |
| Gelo and Koch (2015) [92] | Valuation of community forestry programmes | Ethiopia | 1.24–1.89 |
| Tilahun et al. (2015) [29] | Conservation of frankincense forest | Ethiopia | 5.83–6.42 |
| Amare et al. (2016) [93] | Church forests restoration | Ethiopia | 1.93 |
| Elmi et al. (2016) [94] | Economic valuation for forest conservation for carbon sequestration | Ethiopia | 3.72–6.96 |
| Khuc et al. (2016) [95] | Estimation of urban households' WTP for forest restoration | Vietnam | 24.15 |
| Ramli et al. (2017) [96] | Economic value for the conservation of mangrove forests | Malaysia | 24.15 |
| Solikin (2017) [97] | WTP valuation to avoid deforestation and degradation | Indonesia | 14.48 and (20.25) |
| Ariyo et al. (2018) [98] | Forest conservation | Nigeria | 4.39 |
| Iranah et al. (2018) [31] | WTP visitors' estimation for forest conservation and restoration | Mauritius | 4.28–8.85 |

**Table 16.** *Cont.*

| Authors and Year | Description | Country | Annual Mean WTP Value (USD) |
|---|---|---|---|
| Arabomen et al. (2019) [99] | Economic valuation for urban trees' conservation and environmental services | Nigeria | 16.46 |
| Sardana (2019) [100] | Valuation of tourism restoration of agroforest ecosystems | India | 3.22 |
| Endalew and Wondimagegnhu (2019) [101] | Conservation of church forests | Ethiopia | 7.34 |
| Gordillo et al. (2019) [32] | WTP estimation for forest conservation | Ecuador | 42.95–85.09 |
| Bamwesigye et al. (2020) [71] | WTP estimation for existence value of forests | Uganda | 16.94 |
| Endalew et al. (2020) [102] | Conservation of church forests | Ethiopia | 9.12 |
| Hasan-Basri et al. (2020) [103] | Mangrove forests conservations | Malaysia | 4.90 |
| Khai et al. (2020) [104] | Economic valuation for ecosystem conservation | Vietnam | 49.35 |
| Sharif et al. (2021) [64] | WTP for conservation of recreational forests | Malaysia | 4.48 |
| Truong (2022) [105] | Community perception and participation in forest conservation | Vietnam | 0.014 |
| Kassahun and Taw (2022) [106] | WTP valuation for baobab trees' conservation | Ethiopia | 3.91 |

CVM can be considered the most methodologically sound approach to obtain the economic value of natural and cultural assets, as in the case of cork oak forests [80,107,108]. CVM through the estimation of the monetary value can support DMs in the design of suitable policies and actions for protecting, valorising, and managing cork oak forests, and more so in general, FOWLs, thus contributing to their sustainable forest management (SFM) [109]. Moreover, CVM is regarded to be the only one to calculate the economic value of an asset in all its meanings.

However, some aspects should be taken into account since they may affect the valuation and the precision of the results. For instance, a user could be influenced by the payment option provided by the questionnaire regardless of whether it is considered less reliable; or in the case of an iterative game, the beginning value could impact the final estimation. The presence of outliers could also influence the valuation, for example, the user may condition the results of the research with a different response than the real monetary measure that (s)he would have attributed to the valuation objective (e.g., warm glow); or the user may tend to hide his/her preferences, waiting for other users to state their willingness to pay for the commodity or service that (s)he probably will not use (e.g., free rider).

A careful design of the survey and the research experience are fundamental to design the evaluation scenario and reduce the occurrence of strategic behaviours and outliers.

The evaluation method developed in this study has broad employability in various contexts and with regard to particular geographical issues. The strengths of the method lie in the definition of an agile, but at the same time comprehensive, set of indicators of ecosystem services, which allow a dual evaluation (biophysical and cultural) and, above all, is spatialised by GIS, thus making it useful for planning, territorial, and landscape policies [110]. In fact, this method of evaluation of cultural ecosystem services can explicit their role of "bridging concepts". Ecosystem services are an expression of the widespread awareness of the need to integrate environmental issues into territorial policies, as well as an important tool for the definition, implementation, and communication of sustainability policies, capable of effectively combining conservation and development, thus highlighting the added value that ecosystems provide to society and the economy. This potential is obvi-

ously closely related to the clarification of their evaluation, mapping, communication, and possible 'payment' (PES), at the heart of various research and institutional initiatives [25,26] for the development of large-scale local planning [111–113], with a view to ensure a high level of biodiversity. From a design perspective, the evaluation of ecosystem services is a particularly useful tool for determining the quality of the territory, health, and resilience, and it is essential in order to support the identification of strategic areas for an ecological network, for the development of green and blue infrastructures, as well as to identify landscape, fruitive, and economic values linked to the territories of the waters. The analysis of the ecological network and the optimisation of the improvement of the connectivity value on ecosystem services [36,37]—through innovative processing in terms of remote sensing (3D visualisations and thermographic survey for the evaluation of indicators specifically related to the fire risk)—is therefore a possible in-depth analysis, which can be developed in a forthcoming research activity, supporting the identification of strategic areas for the network, whose potential for strengthening ecological functionality is highlighted. From this perspective, the increase in ecological connectivity is therefore to be understood as the bearer of a multiplicity of values, not only those strictly related to biodiversity, but also to landscape, fruition, and economic values.

## 7. Conclusions

The paper proposed a CVM as an exploratory approach to valuate the WTP of Goceano's cork oak forests of Sardinia (Italy). The relatively simple applicability of the method, which also guided the choice of some estimation methods, responds to the desire to prepare a tool that can be largely used in the context of landscape, regional, and urban planning. These same advantages of the method, spatialisation, and easy applicability evidently also constitute the aspects of partial weaknesses, directing it towards a necessary procedural simplification.

From the perspective of a further development of the research here presented, it is possible to foresee, although not wanting to abandon this approach, an in-depth study of some of the indicators identified, with reference to those of energy use or to the extension of the evaluation of cultural ecosystem services (as we have seen, more difficult to estimate). Furthermore, considering this valuation tool as a potential support for planning and managing policies of the cork oak forest landscapes, it is appropriate that the valuation carried out is integrated with an analysis of the trade-offs [114], thus identifying the potential conflicts and synergies between the multiple functions of cork oak forests (first of all, the economic aspect related to crafts) and effectively supporting the choices of planning and managing territories.

In general, considering the issue of ecosystem services in territorial and landscape planning policies supports planning schemes that are oriented towards a sustainable development perspective in which the act of diversity conservation—not only biological (biodiversity), but also landscape and cultural—is central, thus supporting an interpretation of the forests, as well as through an increase in ecological network and the preservation of its core areas.

However, taking into account this last aspect in relation to the regional landscape plan (RLP) of the region of Sardinia—currently under review to include inland territories as the approved instrument (2006) only concerns coastal areas (integration and extension to the whole territory is in progress, as required by the Italian Cultural Heritage and Landscape Code)—these types of analyses could constitute an effective support for elaborating an articulated and complete analysis of the values of the forest landscapes (not only of the cork oak forests) and, consequently, for declining in an appropriate way, at every level of government of the territory, the protection measures (constraints), management, and planning. This need, however, clashes with the complexity of today's territorial framework: the poorly defined methods for an active safeguard of environmental and landscape resources; the need for the reorganisation of urban transformations; the interpretative uncertainty of the SEA procedures of urban plans; the current incompleteness of the

guidelines for the adaptation of urban plans to the RLP; and the hydrogeological plan (which detail the elements connected to the reorganisation of knowledge but reduced to the mere adaptation of the cartographic drawings)—all of which are aspects that, at the local level, clearly need different tools and implementation strategies that the current revision of the RLP is called upon to consider.

In conclusion, the experimentation of methods and tools for evaluating ecosystem services allow us to bring together the different spheres—biophysical and cultural—that action on the landscape requires to develop "a multilevel planning ( . . . ) through the construction of a supply chain horizontal between responsible subjects, to be pursued from the early stages of elaboration of the regional landscape plan with a concrete participation of local authorities and the use of guide tools for their action adaptable to the specificities of the different landscapes" [115].

**Author Contributions:** Conceptualization, L.L.R., M.C.B. and A.V.; Methodology, L.L.R., M.C.B. and A.V.; Software, V.A. and F.D.; Validation, A.V.; Formal analysis, L.L.R., V.A. and F.D.; Investigation, L.L.R., V.A. and F.D.; Resources, L.L.R.; Data curation, L.L.R.; Writing—original draft, L.L.R.; Writing—review & editing, L.L.R., M.C.B. and A.V.; Visualization, L.L.R.; Supervision, M.C.B. and A.V.; Funding acquisition, A.V. All authors have read and agreed to the published version of the manuscript.

**Funding:** This research was funded by Agenzia Fo.Re.STAS Sardegna (Ex Artt. 3 E 4 Accordo Attuativo Ex Art. 15 Italian Law 241/1990 between Politecnico di Torino, Dipartimento Interateneo di Scienze, Progetto e Politiche del Territorio (Dist) and Agenzia Forestale Regionale per lo Sviluppo del Territorio e dell'ambiente della Sardegna (Forestas) of 27/12/2018).

**Institutional Review Board Statement:** Not applicable.

**Informed Consent Statement:** Not applicable.

**Data Availability Statement:** Not applicable.

**Acknowledgments:** Part of this work has been developed in the research work on the "Valuation of the attractiveness of cork oak forests in Sardinia region" as a joint collaboration between the Interuniversity Department of Regional and Urban Studies and Planning of Polytechnic University of Turin and the Forest Regional Agency of Regione Sardegna (FoReSTAS). We acknowledge Sara Maltoni and Massimo D'Angelo (FoReSTAS) for their support in this research.

**Conflicts of Interest:** The authors declare no conflict of interest.

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
