# Peer review of "A Contingent Valuation-Based Method to Valuate Ecosystem Services for a Proactive Planning and Management of Cork Oak Forests in Sardinia (Italy)"

_sustainability, doi:10.3390/su15107986_

Round 1

Reviewer 1 Report

Dear authors,

Thank you very much for your work on one of the most special ecosystems in Europe. The manuscript is well-written and well-linked to existing literature. Results provide a sound background for policy-makers and an interesting outlook on WTP for ES in Sardegna.

My comments are namely oriented to following aspects:

Characteristics of residents and tourists
Would it be possible to carry out some comparative analysis that would identify differences between those two groups?

Regression model
Most of the independent variables are binary and "very well selected", which results in extremely high, in my opinion, R2 values. I think you need to consider the regression results as supportive of your discussion but not for strict conclusions.

Tables visualisation
I think this is something which could/must be improved. Using CAPS-LOCK, comma for decimals, in case of regression, I would remove intervals and replace p-values with "*" symbols for coefficient (they are all 99% anyway!).

Discussion and conclusion
I see a certain overlap between those two chapters. Text related to the methodology could still be part of the discussion - suggestions and recommendations for following studies.
Nevertheless, the conclusion could be shorter and linked to the aim specified at the end of Chapter 1 (lines 97-99).

Thank you once again for the effort you put in this study.

Author Response

Dear Reviewer,
We very much appreciate your revisions, which we consider really important to improve the quality of our paper.
Below, we try to respond to your suggestions in a punctual manner:

  • Characteristics of resident and tourists: at lines 412 and 419 we have added a small consideration of what we mean for 'residents' and 'tourists'.
  • Regression model:  we considered regression results only as supportive of the discussion
  • Table visualisation: we discussed this with the other authors and decided not to replace the p-value with symbols, so as not to lose the precision of the result.
  • Discussion and conclusion: we have reduced the Conclusions moving some parts to the Discussion section.

Please see the attachment for the revised version of the paper.

Thank you very much for your work

Reviewer 2 Report

Comments to the authors:

The present study describes a model through which a Contingent Valuation Approach to support public bodies in the exploration and assessment of cultural ecosystem services generated by Forest and Wood Lands in cork oak forests in Sardinia region (Italy).

 It needs to be presented in a better manner to understand it.

The manuscript has many errors, in many places it is hard to understand due to poor language, long sentences and lacks in connectivity (synchronization). Hence, there is a scope to improve the quality of the Ms to make it readable to a wide range of audience. The Ms lack the coherence in the text and lacks in the connectivity. This Ms is written causally.

The conclusion section needs to be reduced. The results, conclusion and discussion section needs to be written precisely. I regret to inform you that I cannot accept the article in its current form.

Specific comments:

Title- The title of the article needs to be updated that describes the article.

Abstract- The abstract portion needs to be re-written. In the title it is cork oak forests while in the abstract it is cork oak woods. I suggest use a proper term throughout the text.

Introduction- The introduction section needs to be improved because this portion is not focused. The introduction section lacks citation that support the text. The introduction section is hard to understand what authors are trying concentration? Check it, if the introduction is matches with the objectives of the study? 

L-28- What do you mean by ‘deep fire’?

Just forward 3 main objectives of the paper

Methodology

There is no flow in the method section, I have observed a huge mismatch in the method section. Thus, It is hard to understand method section; how method is implemented, data were collected and analysed. How and what the data/information were collected from the review of the literature. Study area must come before the methods is described.

Results- The results section needs to be improved with a coherence and connectivity. Too much tables.

Discussion- The discussion part is not written well, which needs to be written well to describe the connectivity and importance of the study. I am afraid, the discussion part is lesser than the conclusion section. The results, discussion and conclusion sections needs to be improved.

Conclusion- This part also need to be improved. Reduce it up to a small para. Authors needs to move some part of the conclusion to the discussion section. 

Editorial is needed to improve the language of the Ms.

Author Response

Dear Reviewer,
We very much appreciate your revisions, which we consider really important to improve the quality of our paper.
Below, we try to respond to your suggestions in a punctual manner:

  • Forest/Woods: we decided to use always "Cork oak forests"
  • We tried to improve the introduction, better focusing the objective of the work and the structure of the paper.
  • Study area description has been moved before Methodology
  • Tables: we decided together with the other authors to maintain the structure of the tables and not to reduce their number in order to avoid a loss of information.
  • "Deep fire" has been changed in "Unmanaged fires"
  • Discussion and conclusion: we have reduced the Conclusions moving some parts to the Discussion section.
  • We also thank you for uploading the corrections on a separate file: it was very helpful for us.

Please, see the attachment for the revised version of the paper.

Thank you very much for your work

Reviewer 3 Report

Dear authors,

Thank you for the possibility and pleasure to review your article. Without any doubts, it has a high value for the ecosystem service valuation, and your findings can (and should definitely) be applied in practice.

I am very satisfied by the article in its present form. Still, the only minor issue I'd like to be corrected, is the table captions. Would be great to see the variable descriptions in the captions to the tables 5-10, 12 and 13.

Hope that fixing this issue is a fast and easy task, so this article sees the world soon. 

Author Response

Dear Reviewer,
We very much appreciate your revisions, which we consider really important to improve the quality of our paper.
Below, we try to respond to your suggestions in a punctual manner:

  • The Table 5 reports all the description of the variables (independent and dependent). The variables in the following tables therefore refer to the descriptions in Table 5. For reasons of space, we have therefore decided not to repeat the descriptions of all the variables considered.

Please, see the attachment for the revised version of the paper.

Thank you very much for your work

Round 2

Reviewer 2 Report

L-114- Quercus suber

L-558- Cause of this lack of data, correct this sentence

L-578- Discussion of results, ‘Discussion’

Still too much tables.

Authors correct all such kind of errors in the Ms 

It has improved.

Author Response

Dear reviewer,

we have considered your corrections and suggestions for improving the paper. In particular, we have punctually corrected the marked parts and removed one table, in agreement with the other authors. We have revised the English again in various parts. We hope that these changes will be helpful in proceeding with the publication of the MS. We attach the text with the changes in view.

Thank you for your interest and efforts.
